# Learning with sparse reward in a gap junction network inspired by the insect mushroom body

**Tianqi Wei[1,2], Qinghai Guo[3], Barbara Webb[1]** *

**1** Institute of Perception, Action, and Behaviour, School of Informatics, University of Edinburgh, Edinburgh, United Kingdom, **2** School of Artificial Intelligence, Sun Yat-sen University, Zhuhai, Guangdong, China, **3** Huawei Technologies Co., Ltd., Shenzhen, Guangdong, China

* B.Webb@ed.ac.uk

**Data Availability Statement:** The code supporting the results in this paper is available at: https://github.com/InsectRobotics/DynamicRoutingPublish.git.

## Abstract

Animals can learn in real-life scenarios where rewards are often only available when a goal is achieved. This 'distal' or 'sparse' reward problem remains a challenge for conventional reinforcement learning algorithms. Here we investigate an algorithm for learning in such scenarios, inspired by the possibility that axo-axonal gap junction connections, observed in neural circuits with parallel fibres such as the insect mushroom body, could form a resistive network. In such a network, an active node represents the task state, connections between nodes represent state transitions and their connection to actions, and current flow to a target state can guide decision making. Building on evidence that gap junction weights are adaptive, we propose that experience of a task can modulate the connections to form a graph encoding the task structure. We demonstrate that the approach can be used for efficient reinforcement learning under sparse rewards, and discuss whether it is plausible as an account of the insect mushroom body.

## Author summary

Learning in situations where reward is only rarely encountered is difficult. It is hard to discover the right sequence of actions when most actions, most of the time, provide no apparent progress towards a goal. Inspired by a neural circuit in the insect brain, and using direct electrical connections between neurons as well as synaptic connections, we present a new algorithm for learning. The model represents the states of the world with nodes and an electrical connection between two nodes is strengthened when the two corresponding states occur consecutively. The connections between nodes can also become associated to output actions that correlate with (hence are assumed to cause) transitions between states. When a particular goal is chosen or associated with a reward, for example, the target location in a navigation task, a flow of electrical current through the nodes will find the shortest path from the present state to the goal state and trigger the appropriate actions.

**Funding:** This work was supported in part by the Huawei Technologies Co., Ltd. via grant number YBN2020045132 awarded to BW & TW. The funders had no role in study design, data collection and analysis. QG is an employee of Huawei Technologies Co., Ltd. and participated in preparation of the manuscript and decision to publish.

**Competing interests:** The authors have declared that no competing interests exist.

## Introduction

It is widely accepted that gap junction connections between neurons could play a role in the communication and computations performed in biological nervous systems [1]. A gap junction, in contrast to a chemical synapse, allows direct ion flow from one neuron to another, which has multiple potential consequences. For example, it has been proposed that parallel fibres forming the output from pyramidal cells create an 'axonal plexus' [2] through gap junction connections, such that action potentials in one neuron create spikelets (or even trigger action potentials) in neighbouring axons [3] and thus affect downstream chemical signalling by the neighbours [4]. To date, suggested functional roles of such interactions include synchronization, regulation of oscillations [2, 5], linear interpolation [6] and faster communication [7] between neurons. Here we consider whether gap junctions in an axonal plexus could support more complex computational functions, suitable for encoding experience in a way that would support reinforcement learning. We present an abstracted model to explore this possibility that treats the axonal plexus as a resistor network. A key inspiration for the architecture of this model is the insect mushroom body.

The mushroom body (MB) is an increasingly well-studied circuit in the insect brain [8–10] which plays an important role in learning and cognition [11–13]. The connectome of the MB in larval and adult MB is now well described [8, 10, 14, 15]. Sensory inputs, such as the activity of olfactory receptor neurons (ORN), are mapped via projection neurons (PNs) to a large number of Kenyon cells (KCs). Each KC only attaches to and reads from about one to seven PNs, such that the sensory inputs are mapped to a sparse code in a high-dimensional space [16]. The KCs extend their axons in parallel (in tightly packed bundles) through the lobes of the MB, which are innervated by mushroom body output neurons (MBONs). Each MBON reads out from a large proportion of the KCs and connects to other regions in the insect brain, with different MBON activities linked to the production of different motivated actions such as approach or avoidance [17]. The main substrate for learning appears to be plasticity in the KC-MBON connections, modulated by dopaminergic inputs to the lobes.

Recent research on the mushroom body has identified abundant connections between KCs [10, 14, 15, 18]. For the most part, these KC-KC connections occur between KC axons where they converge onto a single MBON post-synaptic density, sometimes forming 'rosettes' with a set of KC synapses surrounding the MBON [14, 19]. While the connectome provides evidence for chemical synapses between KC axons, dye-coupling experiments have also revealed gap junction connectivity between KCs in the MB lobes [20], both within and between different KC-types. Memory deficits in a visual learning paradigm were observed in this study when gap junctions were blocked. Knock-out of gap junctions in the dorsal paired medial (DPM) and anterior paired lateral (APL) neurons, which both receive multiple KC inputs, impairs anesthesia-sensitive memory in odour-shock learning [21, 22]. More recently, knock-out of the gap-junction gene Inenxin5 or application of gap junction blocker in KCs was shown to affect retrieval of anesthesia-resistant memory [23]. We thus suggest that the KCs in the lobes of the MB, along with the DPM and APL, might form an axonal plexus, such that their interconnectivity plays a crucial role in the downstream activation of the MBONs.

Although the KC-KC connections described so far are both chemical and gap-junction synapses, we focus here on a possible interpretation of the gap junction connectivity. In fact, the model we present is intended to be quite general, but depends on the assumption (which holds for the KCs, but also for many other circuits) that the neural population provides a sparse representation of sensory states. We suggest the axonal plexus of such a neural population can be interpreted as a resistive net, in which the present state is 'pulled high' by the activation of corresponding nodes (e.g. an action potential initiated in one axon), while a target state is 'set

low' by having the corresponding nodes leak current (e.g. an axon with increased ratio of potassium channels or chloride channels). The key consequence of such an interpretation is that the current flow through the whole axonal plexus (from high to low) will be shaped by the relative resistances (strength of the gap junctions) between each pair of axons. As previously established [24] such a resistive net can be used to efficiently generate a shortest route from the current state to the target state if the resistances encode a graph of possible state transitions.

This provides an interesting link to a challenging learning problem from the field of computational reinforcement learning (RL): how an agent can learn to produce a sequence of appropriate actions when only achieving the final goal state results in reinforcement. This 'sparse reward' (or 'distal reward') problem is one that biological systems, including insects, seem capable of solving with relatively little experience, but remains a challenge for RL [25–27]. One class of solution to this problem is to provide a mechanism for latent learning [28], during behavioural exploration (in the absence of direct reinforcement), of the causal structure of the experienced environment. For example, the agent might, in so-called 'model-based RL' [29] explicitly learn a state-action-state transition model that in principle allows it later to plan a route between the current state and a state in which it received the reward. However, a problem with such solutions is that the search process becomes inefficient as the number of states increases.

Our model brings these ideas together, additionally inspired by the demonstration that gap junctions can be heterotypic (i.e., with asymmetric ion flow) and that there can be significant plasticity in these connections [30]. Indeed, a number of factors can dynamically regulate the number of gap junctions that are open or closed, or that are present on the membrane, and activity-dependent depression and potentiation have been demonstrated in invertebrate gap junctions [31–33]. We propose that such adaptation could allow latent learning in a resistive net to encode experience of state transitions, and that the downstream consequences of the resulting current flow (equivalent to the KC axonal plexus effect on MBON activity) could control actions. To complete the analogy to RL learning, we further introduce adaptivity in the mapping from current flow to actions, according to how actions influence the state transitions.

We hence propose in this paper a neural graph architecture that can be used to solve sparse reward RL problems. This model goes well beyond existing any evidence for MB function, or indeed for any known gap junction network, so we have renamed it the 'dynamic routing model' and present it in the following as an abstracted concept, rather than referring to specific MB neuron types. In the discussion we will return to the issue of the biological plausibility of its components. For the present, we focus on testing this model on navigation RL tasks. We show it can learn quickly without any reward, and solve discrete tasks successfully under sparse reward. The current flow through the network can find routes between states from the current state towards a goal, during which sub-goals are found, improving efficiency in solving a task.

## Results

In the dynamic routing model (DRM) (Fig 1) the state of the environment and agent (triangles) is mapped to a set of state nodes (circles) which are interconnected forming a network of resistances that encode possible state transitions. We refer to this part of the DRM as the state network. Target states can be marked in the state network by lowering the potential of state nodes, while the potential of the present state node is pulled up, creating currents in the circuit that indicate possible routes from the present state to the target state. We calculate the flow of current using nodal analysis (see Methods and S1 Text for details). Action nodes, connecting to edges in the state network, receive activation from the current flow between state nodes, and determine the behaviour of the agent. Both the connections between state nodes and the

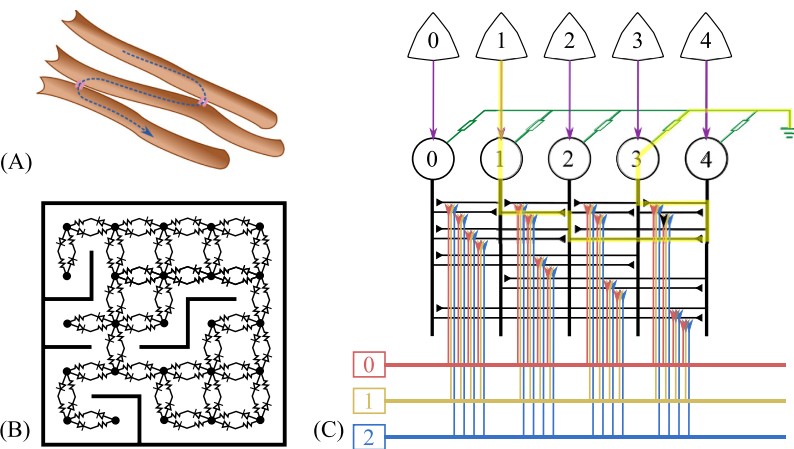

**Fig 1. Overview of the dynamic routing network concept.** (A) Gap junctions allow ions to flow from one neuron to another neuron, which can be represented as asymmetric resistors that connect states (B). (C) State nodes (numbered circles) are connected to each other and ground via gap junctions. A state node is pulled high when the corresponding world state (upper triangle stars) occurs. The yellow highlight marks a current flow from the present state 1 to a target state 3 (connected to the ground) via state 2 and 4. This current flow activates action nodes (rectangles). The connections between state nodes and from state node connections to action nodes are plastic.

connection to action nodes are learned from the exploration of the environment. As described in detail in Methods, the weight of the directed connection between two state nodes is increased if they are experienced on successive time steps, so that the connectivity comes to resemble the environment's state transition structure. Similarly, if such a transition between states occurs after an action was taken, the strength of the connection to that action from the edge connecting these states is increased. Finally, the experience of reinforcement in a particular state leads to an increase in the conductance between that state and ground, setting it as a target for future behaviour.

Three experiments are conducted, two for RL and one for associative learning. The RL tasks are a simple discrete task (Taxi-v3 from OpenAI Gym) and a more complex navigation task in a Voronoi world. We note these two tasks are much more complex than the typical reinforcement learning tasks used in evaluating models of the MB [34–36]. Hence the associative learning task is a discrete state version of a simple associative learning task for direct comparison to insect behaviour. Because several recent models are based on the proposal that MB learning is based on prediction error, we used Q-learning as a baseline for comparison in the RL tasks.

## The taxi domain

The dynamic routing model is first tested in a benchmark RL task, the Taxi-v3 task (Fig 2A) from OpenAI gym. The environment is a 5 × 5 grid world with a taxi navigating the grid. The taxi can move on the grid by moving south, moving north, moving east and moving west, and can pick up or drop off a passenger. There are only four locations where a passenger can appear and wait for picking up by the taxi. The destination of the passenger is in one of these four locations. In practice, the information returned from the simulation is not these details but just an integer from 0 to 499 corresponding to a unique combination of the given circumstances. There are three factors that determine the dimensions of the state space, 1) the location of the car (with 5x5 = 25 dimensions), 2) the location of the destination (with 4 dimensions), 3) the location of the passenger (either at a pickup location or in the taxi,

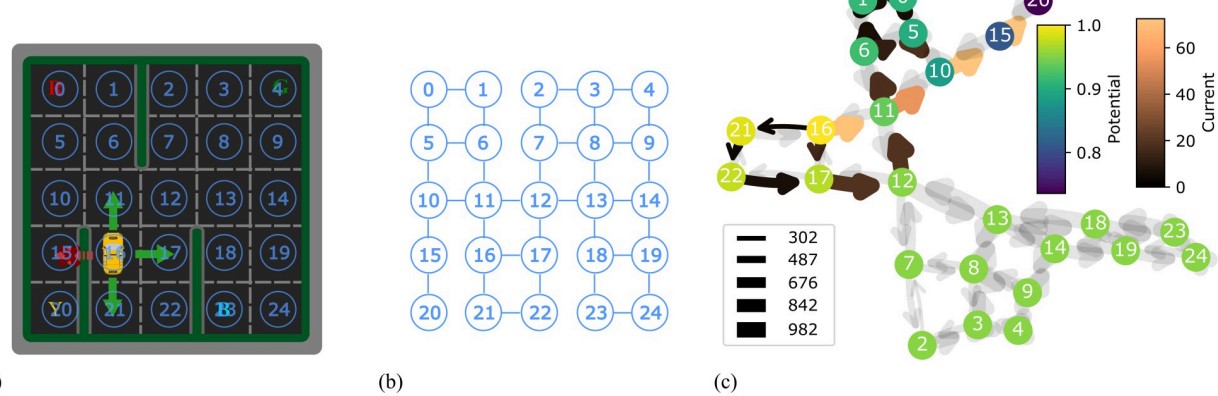

(a)                                        (b)                                        (c)

**Fig 2. The state of taxi locations.** (A) There are 25 locations in the environment, here we number them from 0 to 24. (B) Because the car can only move to adjacent locations without obstacles on its way, the possible transitions between the locations are constrained. (C) Our model learns the possible transitions between locations after training. The conductance strength and direction between nodes are shown by arrow width and direction. The state nodes with high potential (e.g. the current state in the example, 16) are in yellow and the state nodes with low potential (e.g. the goal state, 20) are in purple. The resulting current strength is shown by the brightness of the arrow, in this case creating flow from 16 to 20, guiding the car to take this route.

contributing 4+1 = 5 dimensions). Hence, in total, there are $25 \times 4 \times 5 = 500$ states. The transitions between states are constrained, 1) the car can only move to adjacent locations and cannot move through walls, 2) for an episode, there is only one destination, so there is no connection between the states with different destinations, and 3) the passenger can be picked up or dropped down on the four possible destination-locations, so in an episode, there is only one state connecting the states when the passenger is in the car and the states when the passenger is at a specific destination-location.

Note there are only 400 states reachable during the task, because once the car drops the passenger at the correct destination, the episode terminates, and the taxi cannot reach any other location.

The default training configuration for this task is typical for RL tasks in that it sets a limited number of steps for each episode, and provides (small negative) reinforcement for each step in addition to the large positive reinforcement on achieving the goal state. If the maximum step is reached before the goal state, the episode will be forced to end, and restarted with the agent in a new random initial state. This helps the agent to escape local minima and have a more global sampling. The step reward is necessary for most RL algorithms, including Q-learning, to enable efficient routes to the goal to be discovered.

Our experiment used a different training configuration. In a real-life scenario, a task cannot be restarted easily as a simulated environment, and reinforcement is usually provided only when a goal is achieved. Thus, we configure Taxi-v3 without any limit of step number, and without any intermediate reward. An episode ends only when the target is achieved and the only reinforcement in the episode is at this final step. This makes the problem significantly more challenging and (as we show) it cannot be solved by standard methods such as Q-learning.

**Learning a state network.** To illustrate how a state transition network can be formed from the learning rules in Eq 10. we first train the model to reach a target location using only the information of the taxi location, which is an integer from 0 to 24, as it explores the environment. Fig 2A shows the integers associated with the locations on the grid map, Fig 2B shows the topology of the grid map, and Fig 2C shows the learned state network after exploration of

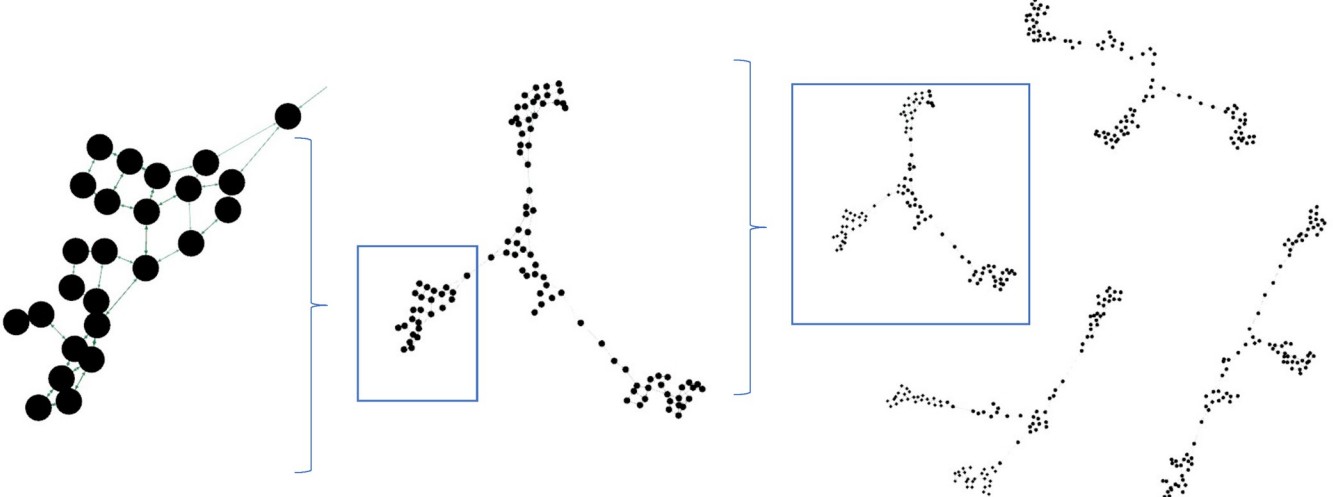

**Fig 3. The learned state network of the full taxi domain task.** This creates four disconnected graphs (one for each destination, which is unique to an episode) each consisting of four subgraphs, for four possible locations of the passenger (the fifth location of the passenger, at their destination, ends the episode, so is not included). Each subgraph represents the topology of locations in the environment.

the environment which has generated a sequence of integer inputs (state changes) which has altered the connection weights in the model. Ignoring the very weak default connections, it has the same topology as the topology of the grid map. That is, the model learns strong gap junction weights between nodes that correspond to possible state transitions in the simulated environment, and weak gap junction weights elsewhere.

We next trained the model in the full state space, i.e. including taxi and passenger locations. Note that there is no separate exploration phase, the model is simultaneously updating the the strength of the state connections and using them to solve the task. The training results in a more complex topology (Fig 3). Because a destination does not change during an episode, the states with different destinations are not connected, forming four graphs, as shown in Fig 3 (right). For states with the same destination, the passenger has four possible locations: three on the map and one in the taxi. The states with a passenger at the correct destination cause the termination of the task, so they are never depicted. Thus, there are four subgraphs in each of the graphs, and each of the subgraphs reproduces the same topology of the grid world map (Fig 3 (left)). The subgraph of the states with a passenger in the taxi connects to the other three subgraphs by the action of picking up or dropping off the passenger (Fig 3 (middle)).

**Task performance.** Our model is trained using only a final step reward of 20 when the taxi drops off the passenger at the correct destination. However, to compare the performance with previous approaches, we calculate an accumulated reward which includes a -10 reward if the taxi tries to pick up or drop off the passenger at locations other than the four destinations, and otherwise -1 reward per step.

Our model converged quickly and achieved optimal results within a few hundred episodes and 20000 total steps, obtaining an average reward of around -5 at the end of the training (Fig 4).

This performance is comparable with the original results for this task presented in the work by Dietterich [37]. In their experiments, the step-wise reward scheme described above was used, and frequent resets were permitted, which is the typical configuration in reinforcement learning tasks and makes learning easier. The convergence time for our algorithm is similar to

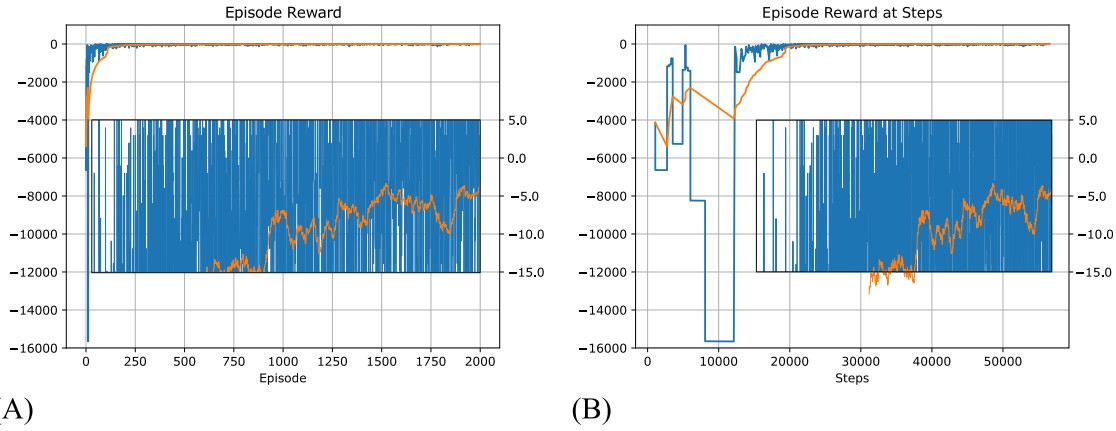

**Fig 4. Total reward during each episode in Taxi-v3 task for the dynamic routing model, using an infinite step limit per episode.** The reward here is calculated with the inclusion of negative reward per step, although only the positive reward at the final step is used in training the model. Blue line: Episode reward. Yellow line: 100 episode average reward. (A) Episode reward per episode. (B) Episode reward (same data as A) but plotted against the steps making up each episode (which differ in duration) to show how reward changes with time. Inset plots are zoomed in regions (changed y-axis) of outer plots, showing how the reward level stabilises around -5.

that achieved using hierarchical RL, in which the task and value function need to be decomposed into sub-problems in advance.

Testing Q-learning under the same configuration as our model (maximum 100000 steps for each episode, and only reward at the final step) we observe that it does not converge (Fig 5) and results in oscillating episode reward with a mean around -40000, and the number of steps in an episode frequently reaches the limit. As a comparison, our model quickly converges to around 20 steps per episode to obtain the goal, as shown in Fig 6.

## Navigation in a Voronoi world

The Taxi-V3 task is a highly simplified version of a real scenario where a taxi picks up a customer and takes them to a destination. In reality, 1) the world is not a perfect grid, even in a city, e.g. there are irregular blocks where the roads intersect in more complex ways; 2) the passengers

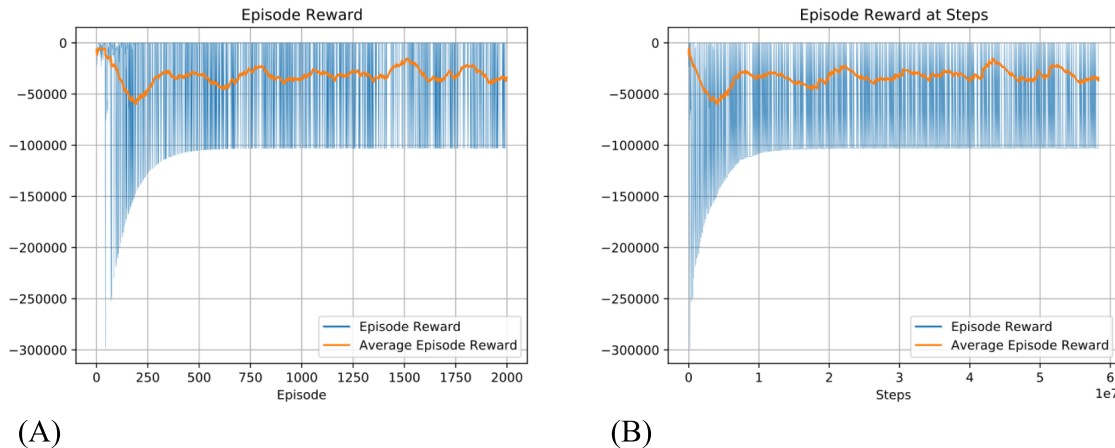

**Fig 5. Result of using Q-learning with a similar training configuration to that used for our model, i.e., maximum 100000 steps for each episode and sparse reward.** Blue line: Episode reward. Yellow line: 100 episode average reward. The Left shows the reward per episode and the right reward per step. Please note the y-axis is not in the same scale with Fig 4. The average episode reward suggests that the Q-learning's performance decreased in the early episodes of training and failed to converge.

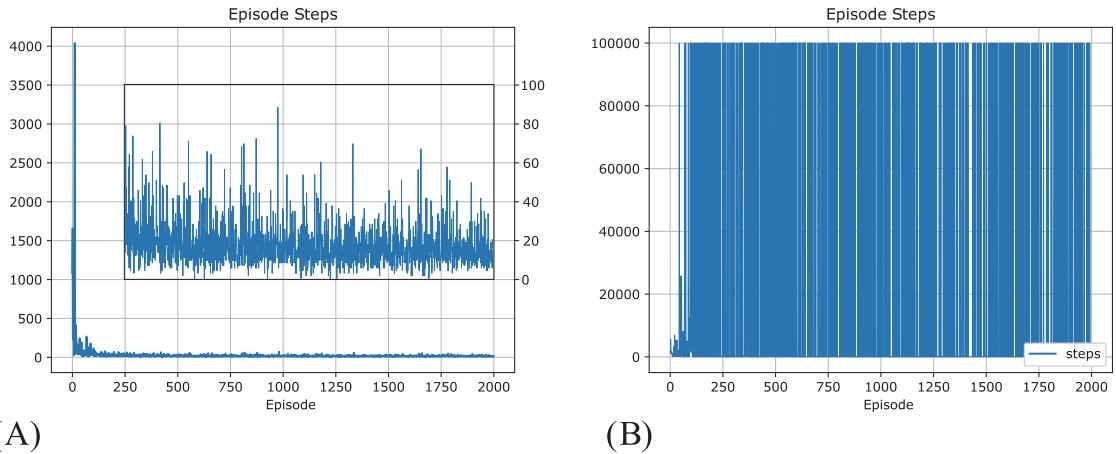

**Fig 6. Number of steps per episode in Taxi-v3 task (A) is the number of steps with the dynamic routing model.** The inset figure is a zoomed-in version of the outer plot showing convergence to around 20 steps. (B) is the number of steps taken per episode by Q-learning in the same training configuration same as Fig 5. Q-learning does not converge.

could appear at arbitrary locations and have arbitrary destinations, unlike the limited four locations in the Taxi tasks; 3) the types and number of actions available for different states vary.

To capture these complexities, we proposed a task with a more realistic topology for navigation generated from the Voronoi diagram, and named it the Voronoi world. In this environment, the agent starts at arbitrary location in a 2D space, and an arbitrary destination is provided in each episode. There are a different number of actions available in different locations depending on the number of neighbouring locations, which varies. The locations (yellow dots in Fig 7) in the Voronoi world could be created to correspond to a real world task, e.g., to represent landmarks such as furniture in an indoor space, buildings or city blocks in an outdoor space. For our simulation, they are generated with Poisson disk sampling [38] which ensures that the generated locations are not too close to each other. A model that performs well on this task should also work in real-world scenarios with similarly complex topology.

Given the set of locations, a Voronoi diagram (Fig 7A, blue) provides a division of the plane based on the locations. The division associates each location with a region in which points are closer to this location than any other location. The Voronoi diagram is commonly used in tasks such as finding a region nearest to a metro station or path planning to maximise the distance from robot to multiple obstacles. In the plane, edges connecting neighbouring locations form the Delaunay triangulation (Fig 7A, yellow). Every edge in the Delaunay triangulation is perpendicular to a corresponding wall in the Voronoi diagram. To create the Voronoi world for the navigation task, a random selection of walls are removed. Correspondingly, every edge that crosses a remaining wall is removed, leaving only those edges on which the an agent can move between locations (Fig 7B). These passable paths form the graph in the Voronoi task.

The state observation provided to the agent by the environment in this task is the current location of the agent and the target location for each episode. Both of the locations are provided as indices as shown in Fig 8A. The actions are discrete and the number of actions the agent can take depends on the number of paths from its location to its neighbours. For example, if there are three paths, then the agent has actions 1, 2 and 3. If there are six paths, then the agent has actions 1, 2, 3, 4, 5, and 6. The correspondence between paths and actions depends on the order of the paths saved in a list associated with a location. When the agent reaches the target location, the episode finishes. When a new episode starts, the current location is given by the end of the previous episode, and the target location is reset randomly. In this way, we

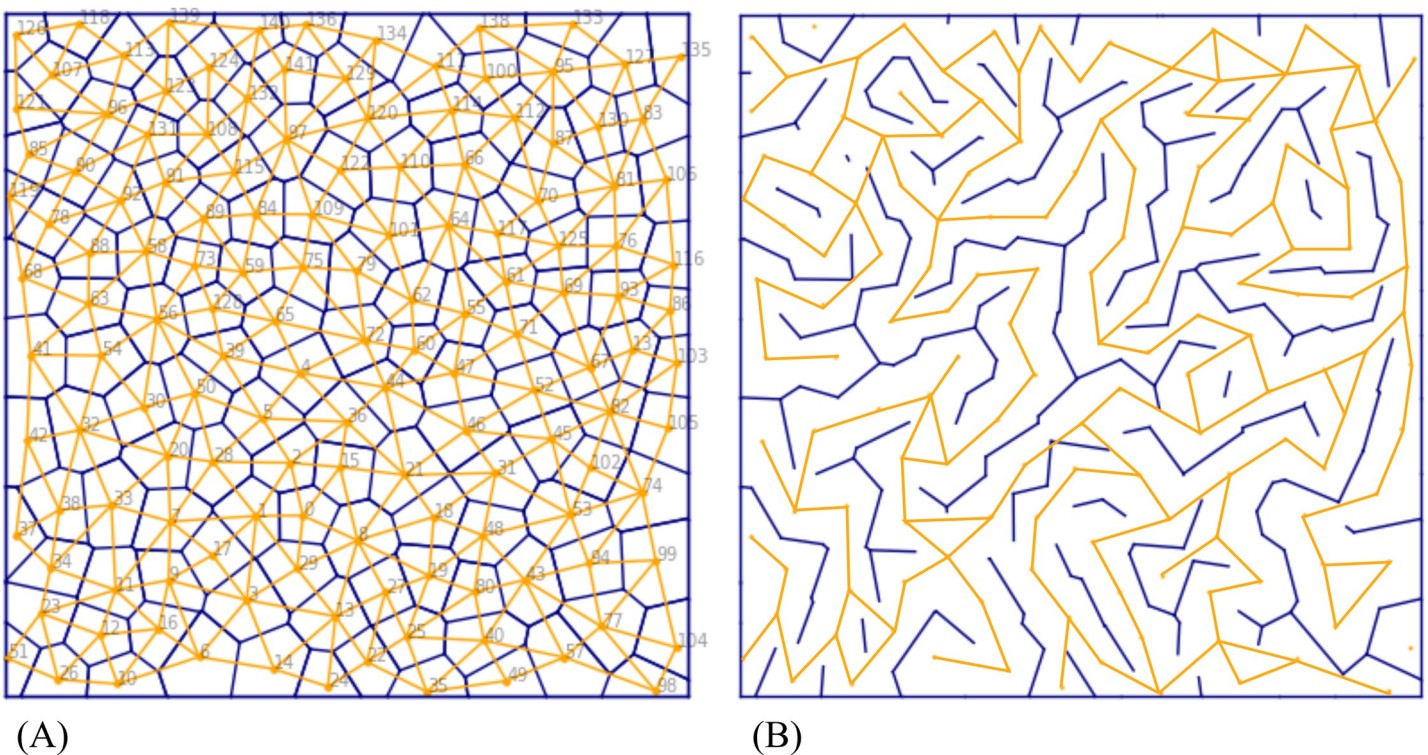

(A)                              (B)

**Fig 7. The Voronoi world task.** (A) Locations are marked using yellow dots and numbered. The corresponding Voronoi diagram is in blue, and corresponding the Delaunay triangulation is in yellow. (B) A maze generated by removing randomly selected walls from the Voronoi diagram.

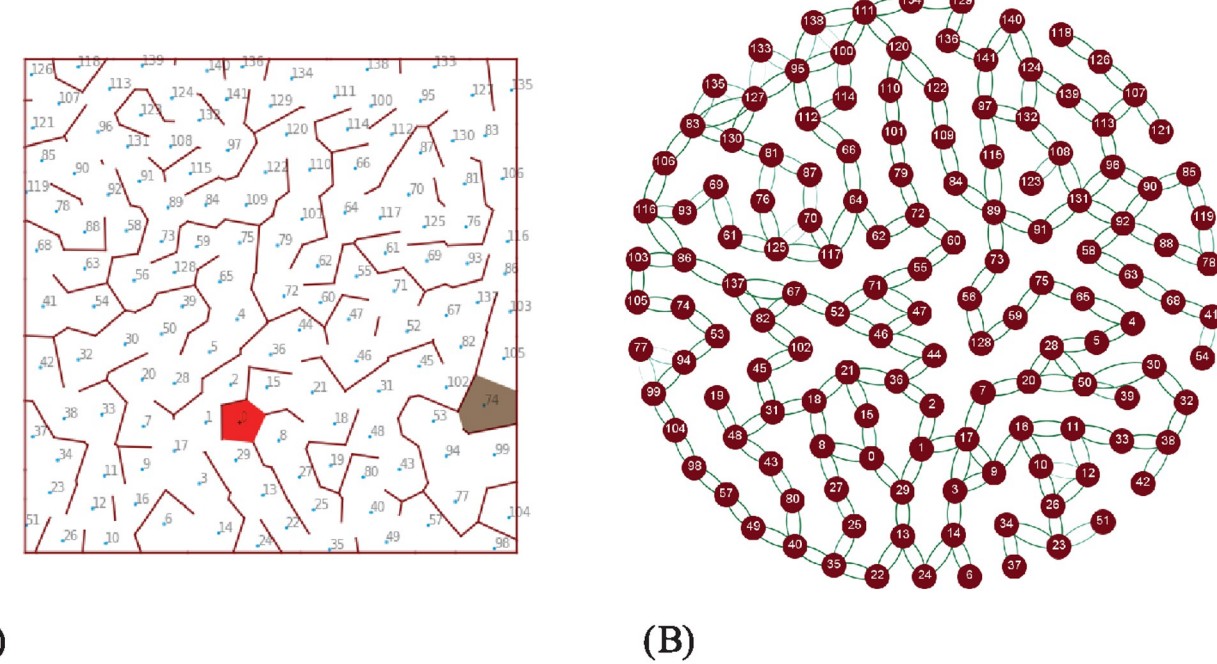

(A)                              (B)

**Fig 8.** (A) The rendered image of the task. The region coloured in red is the goal location. (B) The learned state network for the Voronoi world task. The darker the edge colour, the stronger the connections. Connections with weights below a threshold are not shown.

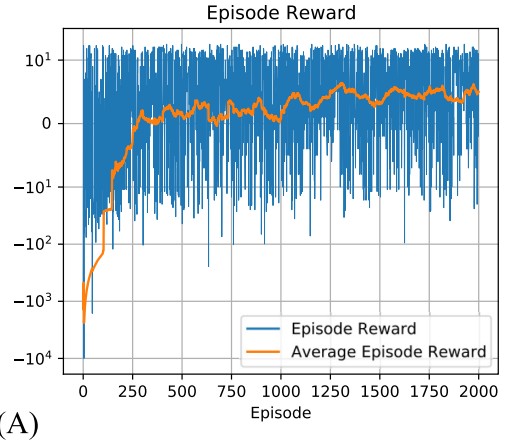
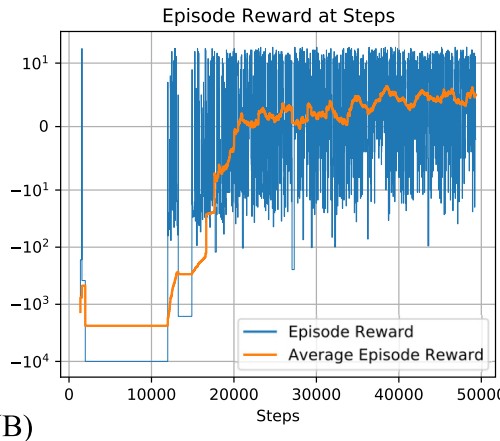

**Fig 9. Cumulative episode reward in the Voronoi world task by the dynamic routing model.** 10000 step limit. Only the reward at the final step is fed to the model. Blue line: Episode reward. Yellow line: 100 episode average reward. (A) is reward per episode, (B) is reward per step. The y-axes are in linear-scale between -10 to 10, but log-scale out of this range.

can simulate a taxi continuously navigating to different destinations, or a robot continuously navigating from one state to another state across the episodes.

To apply our model Voronoi world task, it sets the observed agent location as the activated state neuron and pulls down the state neuron corresponding to the target location to set the target. Then the current from the present state to the target state guides the exploration and learning. We also observed that it is not necessary to recalculate the current flow after every state change (i.e. to update it as the agent moves towards the target) to obtain good results. The field of current set up in the network from the original state to the target can still provide valid guidance for choosing actions even if the state node with the highest potential does not represent the present state. This allows more efficient execution, although it may sometimes produce a less optimal solution.

Using this method, the topology of the Voronoi world is learned, and Fig 8B shows how the topology is represented in the state network. After learning, the topology of the state network reproduces the topology of the Delaunay triangulation of the Voronoi world. The performance in solving the task is shown in Fig 9. To show the improvement over episodes, we calculate the reward per episode by assuming when the agent moves a step, there is a -1 reward for the cost of energy (but this is not used in training) and when the agent reaches the target, there is a +20 reward. The model converges quickly in about 250 episodes or 25000 steps.

Again, we applied Q-learning to the task with a similar configuration, that is, each episode has a large step limit (10000), and rewards are only provided to Q-learning when the target is achieved. Under this configuration, Q-learning cannot converge, as shown in Fig 10A and 10B, the episode reward oscillates strongly, and because the lowest bound on episode reward is fixed, the average episode reward oscillates around -2000, and the number of episode steps frequently reaches the limit.

We also tested our model in the same Voronoi world but changing the number of goals from one to two, to examine whether the model can learn and perform appropriately with more than one reward source. In the task, the two reward sources were reset in every episode, and once the agent reached one of the reward sources, the episode ended. The maximum allowed number of steps in an episode is 10000. Our model is also able to learn this task quickly, reaching -10 in 214 episodes and converging around 0.

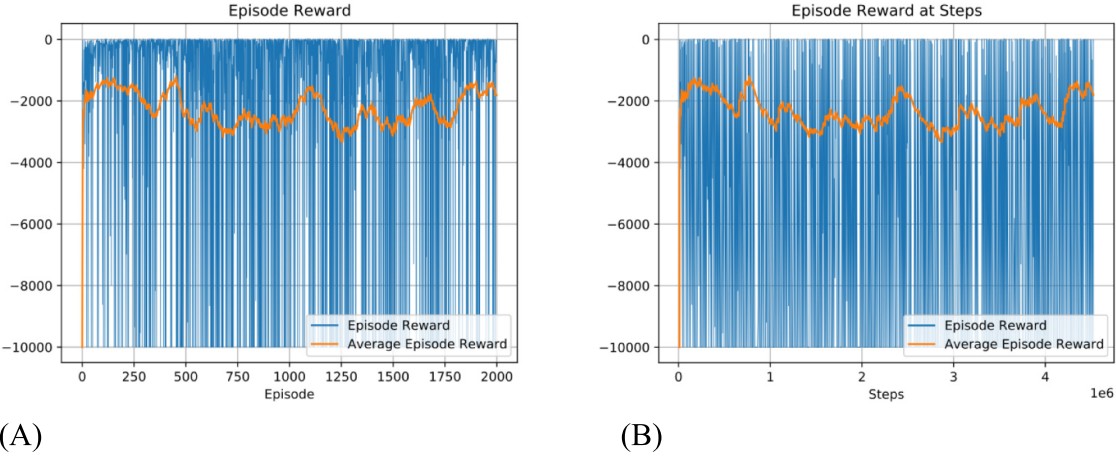

(A)                                                                          (B)

**Fig 10. Result of using Q-learning with a similar training configuration to solve Voronoi World.** That is, maximum 10000 steps for each episode and sparse reward. Q-learning did not converge in such a training configuration. (A) is reward per episode, (B) is reward per step.

### Simple associative learning

Although above we have tested our model in conventional RL tasks, it is also relevant, given our inspiration from the MB, to test whether the model can replicate associate learning behaviour in the type of tasks used to assess larval and adult Drosophila learning. Hence, we implemented a reinforcement learning environment following a standard olfactory associative learning paradigm [39]. In the experiment, naive maggots are trained by experiencing an odour paired with reward, and a second odour without reward, then they are presented with two odours on different sides of a Petri dish. Their odour preference is assessed by the number of maggots on each side after a short test interval.

To represent this experiment protocol in a simplified discrete state space, we implement virtual linear Petri dishes (Table 1) with five locations (Fig 11). At each time step the maggot is in one location. In training conditions, all locations contain the same odour (either amylacetate (AM) or 1-octanol (OCT)). In the test condition, AM is always in the two leftmost locations and OCT is always in the two rightmost locations and the middle location has a mixture of odours. In the two odour case only, there are assumed to be odour gradients, i.e. AM decreases in strength (and OCT increases) from left to right, across the whole dish. The petri dish is also assumed to have a substrate which contains (uniformly in all locations) either fructose (F, a reward), or nothing (N).

At each time step the agent takes one of five possible actions: not moving, making an appetitive action towards AM (or OCT), i.e. moving up the gradient, or aversive action away from

**Table 1. The odours and reinforcers in Petri dishes.**

| Petri dish name | Odour | Reinforcer |
|---|---|---|
| AAN | AM | None |
| AAF | AM | fructose |
| OON | OCT | None |
| OOF | OCT | fructose |
| AON | AM, OCT | None |
| AOF | AM, OCT | fructose |

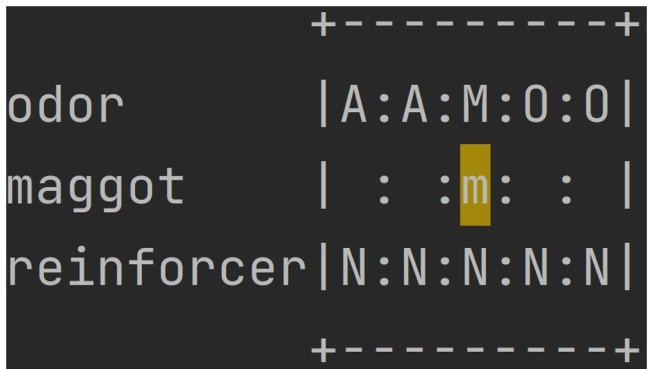

**Fig 11. An example Petri dish ("AON").** A represents amylacetate, O represents 1-octanol, M represents a mix of odours, N specifies no reinforcer.

AM (or OCT), i.e. moving down the gradient (Table 2). If there is no odour gradient (only one odour in the dish) the direction of appetitive and aversive actions is random. If there is reinforcer present (F), and the agent makes an appetitive action towards an odour that brings it into a location with that odour (either through moving up-gradient or randomly), it is assumed to have consumed some of the substrate in that location on that time-step and thus experienced the reinforcer.

The agent has five nodes representing the perceptual state (Table 3) of which only one is active at any time step. If the agent has just taken an appetitive action and experienced a reinforcer, the corresponding state node (4) will be active. Otherwise, if both odours are present in the current location state node 3 is activated, or if only one odour the corresponding state node is activated (2 or 1). A location with no odour would activate the default node (0), but note in the currently described paradigms this situation does not occur.

When our maggot model is in a Petri dish, for example, Petri dish AAF, it perceives AM and state node 1 is activated, then if it makes an appetitive action to AM, it perceives fructose and state 4 activates, causing the edge from 1 to 4 to be learned and associated with action 1, and state node 4 to become associated with reward (connected to ground). For any other action, the maggot continues to perceive AM only. If our maggot model is in Petri dish OON, it only perceives OCT and state 2 will be active.

There are two types of training (Table 4), and each can be followed by testing either with or without the reinforcer, making 4 experimental protocols (Table 5). We used 30 naive maggot models for each of the protocols, and the maggots stayed in each Petri dish for 40 steps.

We counted how many maggots were on each side of the test Petri dishes at every step in the testing showing the preference and learning indexes. At each step, the preference index is:

$$PREF = \frac{\#_{AM} - \#_{OCT}}{\#_{TOTAL}} \tag{1}$$

**Table 2. Action nodes.**

| Action node No. | Action |
|---|---|
| 0 | None |
| 1 | Appetitive AM |
| 2 | Aversive AM |
| 3 | Appetitive OCT |
| 4 | Aversive OCT |

**Table 3. The activated state node given a perception.**

| Perception | Fructose | OCT and AM | OCT | AM | Default |
|---|---|---|---|---|---|
| State node | 4 | 3 | 2 | 1 | 0 |

where $\#_{AM}$ is the number of maggots on the AM side, $\#_{OCT}$ is the number of maggots at the OCT side, $\#_{TOTAL}$ is total number of maggots in the protocol. The learning index combines preference scores from training with one odour vs. the other paired with the reinforcer, to control for innate bias (in real maggots):

$$LI = \frac{PREF_{AM+/OCT} - PREF_{AM/OCT+}}{2} \tag{2}$$

Positive *LI* indicates appetitive memory and negative *LI* indicates aversive memory.

The experiment shows our model can qualitatively replicate associative learning behaviour with positive reward. When the maggots trained with fructose are presented with an odour and tested in a Petri dish without reinforcer, they have the highest learning index. If maggots experience the same training but are tested with fructose, their learning index is lower. As show in Fig 12 this qualitatively matches the data in [39].

Here we compare protocols 1 (train AM+/OCT, test AON) and 3 (train AM+/OCT, test AOF) to explain what happens (see also S1 and S2 Figs). When the maggot model was tested in Petri dish AON, it was put into the middle location, and perceived the mixture of AM and OCT, and state node 3 was activated, and a random action was chosen. If it chose action 1, i.e., appetitive to AM, it moved toward AM and perceived AM only, and state 1 was activated. In this case, the connection that had been formed in training between state 1 and state 4 (fructose) meant there was a strong current guiding the maggot to choose action 1 again to get fructose. However, because the fructose was not in the Petri dish, the maggot kept choosing action 1 more than any other actions, and stayed on AM side. The failure of the action to lead to state 4 would start to weaken the connection to this action, but it would remain stronger than any other actions from state 1. If the maggot was tested in Petri dish AOF, taking an approach action to either odour results in perceived fructose, and state node 4 is activated with a high potential. Because state node 4 is also the target state, there was no strong current flow through the state network to guide the next choice of action. Furthermore, the maggot continued

**Table 4. The sequence of Petri dishes used during training.** Note each protocol pairs one odour with either reward or punishment, and the other with no reinforcer, alternating 3 times between these conditions.

| Training name | Petri dish sequence |
|---|---|
| AM+/OCT | AAF, OON, AAF, OON, AAF, OON |
| AM/OCT+ | OOF, AAN, OOF, AAN, OOF, AAN |

**Table 5. Protocol for training and testing.**

| Protocol No. | Training | Testing |
|---|---|---|
| 1 | AM+/OCT | AON |
| 2 | AM/OCT+ | AON |
| 3 | AM+/OCT | AOF |
| 4 | AM/OCT+ | AOF |

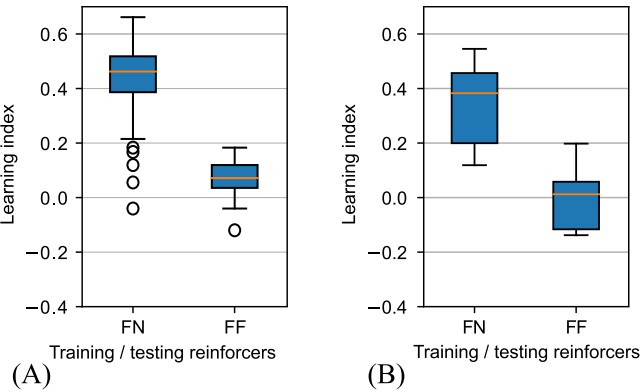

**Fig 12. Boxplots of maggot learning index at every step in the testing.** (A) The learning index with our model in the task. (B) The results in [39]. FN: Trained with fructose and tested without reinforcer. FF: Trained and tested with fructose.

learning during the testing, so in Petri dish AOF, it would learn the state transition from OCT to fructose, and this action (appetitive towards OCT) would start to compete with the appetitive action to AM. Hence overall the time spent in AM would be reduced.

This simple example provides an interesting insight into the question of what is actually learned in associative learning [40]. From a reinforcement learning perspective, it is assumed the agent comes to associate the value of the outcome (fructose or quinine) with the action (approach or avoid) taken in a particular state (in which odour is present). In most MB models, it is instead assumed that the agent associates the value of the outcome (fructose or quinine) with the state (which odour is present) leading it to express an appropriate innate action (approach or avoid) to that odour. In our model, the agent learns which actions lead from one state to another, but value (setting a low or high potential) is associated only with the state in which reward occurs, and dynamically propagates through the network to control action. We note this means the model could potentially account for devaluation phenomena [41].

## Discussion

The model proposed in this paper was inspired by the newly discovered gap junction network between parallel Kenyon cell axons in the insect mushroom body. We suggest this can be interpreted as a graph that encodes, through experience, the causality between actions and states. By treating the gap junctions as rectified resistors, current flow from the previous or current state can automatically find the shortest path through the graph to a goal. We have implemented this concept in a computational model and show that it can perform better than standard RL methods in several benchmark tasks, most notably, under realistic conditions of truly sparse reward (when most states have no reward) and unlimited episode duration.

### Learning under sparse reward

There are a number of existing approaches to improve reinforcement learning (RL) under sparse rewards. One broad category is curiosity-driven learning [25, 26, 42, 43], which assumes there exist internal rewards related to seeking information in addition to the external reward. Internal rewards can be determined by the count of visited states or the variation/error to predict the next state given an action, for example. In latent space exploration [27] the intrinsic reward of curiosity is not computed according to explicit prediction, but using a latent space

that codes the features of inputs, which is more robust than the raw observations. Both approaches can generate internal rewards, and with these rewards, agents can use existing RL models for learning. However, these approaches need to introduce additional models for exploration or computing rewards, which introduces extra computation and training costs. They might be considered special cases of introducing an auxiliary task, that is, an additional cost-function related to the main task that provides a more continuous learning signal, e.g. learning depth prediction in a navigation task as additional auxiliary tasks [44]. In contrast, our model uses the currents to guide the agent to the target, and also uses the same currents to find the bottleneck and guide the agent's exploration of actions to approach and pass through the bottleneck.

Another broad class of methods involve using data collected during exploration to learn with respect to virtual goals that differ from the task goal. For example, in hindsight experience replay (HER) [45], exploratory trajectories that do not lead to the reward state are 'replayed' with the final state actually achieved set as the goal, using an off-policy RL algorithm, to gain information about the observed state transitions.

Alternatively, within model-based RL, experience can be explicitly used to learn a model of the task dynamics that allows planning to be integrated into the learning and acting loop. Such models can be exact (explicitly representing all experienced or all possible state transitions) or approximate, with a wide variety of function approximation methods having been explored to date [29]. Separating learning of the task dynamics and the reward structure is also a key concept for the successor representation approach to RL [46, 47].

The approach described here resembles an exact tabular model [48] in that it learns the connections (transitions) between discrete states according to the number of times this transition is experienced. However, note that our current model learns state-state transitions and then associates actions to these, rather than learning (state, action)-state transitions. This reduces the size of the transition matrix but also enables generalisation to learning in conditions where the actions of the agent are not the sole determinant of state changes, e.g., learning from observation (not tested here). It separates, conceptually, learning about the rewarded states in the environment from the cost of actions. Also, it does not learn or store value as a global solution, but only associates it explicitly with rewarded states. Then, during the action loop, it effectively interpolates that value via the current flow in the network, finding the shortest path and activating the relevant sequence of actions.

## Using a resistive network for planning

A potential advantage of our method is that it is easily parallelised and could be implemented in analog hardware so that the current flow is computed physically. This aspect of our model relates to previous work on resistive networks [24] for analog parallel computing. An early use of this concept was as acceleration for Laplacian Operator to find edges in an image [49]. Later, it was developed for hardware-accelerated path planning, particularly as an improvement over potential field methods, and applied to scenarios such as mazes [24, 50–53], grid worlds with obstacles [54, 55], city roads [56], and robot arms [57, 58]. The platforms for the computation include Field Programmable Analog Arrays [52], Very Large-scale Integration [24, 51, 59], and circuit simulators such as PSPICE [55]. A range of equivalent physical implementations of the principle have been suggested [60], such as using a thermal camera to image the current flow in a graph instantiated as a printed circuit board [61].

The application of resistive grids for planning in the examples above typically assumes the states correspond to regularly distributed locations in 2D or 3D space, with connections only possible between adjacent states. The network is predefined or computed according to a

description of the space, with symmetric connections. Here we use asymmetric connections, extending the approach to capture scenarios with asymmetric (irreversible) state transitions, such as navigation on one-way roads or consumption of power. And our model is also novel in that in principle it allows the connection of any two states, not only predefined neighbours, with the connections generated by learning, to represent causation between abstract states rather than locations.

The concept of current flow in this method might seem to resemble methods that are equivalent to particles or 'activity' diffusing through a graph. However, it differs in that: (1) the currents are hard to compress, whereas flows formed by particles can change their concentration, which alters the dynamics; using current the solution can be found without simulating the flow by nodal analysis. (2) the grounded state node attracts currents, whereas flow formed by particles is only pushed by the source that releases the particles; (3) this method takes the current strength as an important factor in choosing actions, whereas the methods with particle flow pay more attention to the concentration of particles. Hence, for example, if there is a weak bridge connecting two sub-graphs which contain the present state and target state, respectively, the power of the current in our method can concentrate on the bridge, whereas particle diffusing methods might not distinguish the node connecting the bridge from other nodes.

## Biological plausibility

Learning of state, or more explicitly, place, connections to form a topological graph has been investigated in a range of hippocampus-inspired models, often linked with activity propagation methods for planning, e.g. [62–65]. A similar model that uses 'virtual odours' in a learned graph [66] to move towards goals draws a comparison to the mushroom body architecture, and similarly suggests that recurrent KC connectivity could form the substrate for the connection between graph nodes. The mechanism presented here is shown to be effective when applied to a challenging RL problem, learning under sparse reward. But is it biologically plausible? We believe it illustrates how an axonal plexus—combined with directional, adaptive gap junctions—could support more complex computational functions than has been generally assumed to date, such as learning and planning across sequences of sparsely encoded sensory states.

On the other hand, there is no direct evidence to date that KC-KC gap junction connections can be modified in the way we propose, although several authors have speculated that gap junction modulation could play a role in associative learning in the MB [23, 67]. Indeed, the functional relevance and properties of KC-KC connections is an open question, as the extent of these connections has only recently been recognised, and there is still some debate as to the extent of gap junction vs. chemical synapse connectivity [68]. Axo-axonic chemical synapses are an alternative option for learning of spatiotemporal patterns [19] or sequences of sensory states [69]. Either way, we consider it plausible that connectivity between KCs could play a role in modulating the activity of MBONs, such that MBONs (and the actions they control) can effectively become associated with transitions between states. Specifically, we suggest that if an MBON requires simultaneous or consecutive input from two (or more) KC synapses to fire, then this could be enhanced if axo-axonic connections from the first KC can act to generate transmitter release from the second. Such a mechanism would remain consistent with the well-evidenced adaptive changes in connectivity within the insect mushroom body by which KC to MBON synaptic strength is altered under the guidance of reinforcement signals from the DANs [10], such that specific patterns of KC input become connected to specific actions. With regard to directed current flow across an axonal plexus that could resemble a resistive net, an action potential in one KC could correspond to 'pulling up' this node, but less clear (for

KCs or axons in general) what would constitute 'pulling down' to create a target for current flow. In this regard an intriguing possibility is suggested by the observation that inhibition of KCs by the (non-spiking) APL can be highly localised and axon-specific, with KCs causing stronger inhibition on themselves if more strongly connected to the APL [70]. More generally, any mechanism by which the target axon becomes leaky to current could potentially play this role. Perhaps the least plausible assumption of our model is that each node can (potentially) form or strengthen a connection to any other node, whereas the connectivity of KCs, although extensive, is not all to all. Relaxing this assumption would set some "innate" constraints on what state transitions can or cannot be learned, which might be considered biologically plausible. Further, this assumption may become less critical if we also relax the assumption that each state in the world maps to only one node, as discussed next.

### Multi-node activation

Currently, the model uses only one state node to represent the state of a task at a given time, which is different from the coding of sensory input in the MB. In the MB, a small proportion of KCs will be activated together forming a sparse code, which is more efficient than our one-hot style coding. In principle, our model could be extended to deal with sparse coding:

1. Edges would be strengthened according to the negative correlation of state node activities.

2. Targets could be set by grounding multiple state nodes.

3. Action nodes could be activated according to multiple edges from the presently active state nodes.

In Fig 13 we show an example of simulating current flow when a group of state nodes are active and another group represent the target, given a predefined set of connections between the nodes, showing that the resistive net architecture will still find a suitable path. The groups representing each state need not be independent but can overlap to represent how different dimensions of task states can change asynchronously. In future work, we will explore learning and task execution with sparse activation to determine if there are critical constraints on proportion active and overlap to enable learning of the task. It is interesting in this context to note that there are several algorithms that interpret the input circuits of the MB as locality-sensitive hashing [71] by which high-dimensional input can be mapped into the state represented as a sparse number of active KCs.

### Other extensions and future work

Although the current model is applied to discrete tasks in the experiments in this paper, it is intrinsically continuous and can be applied to continuous tasks with a few modifications. For example, the retrieval of action and subgoals does not rely on any discrete process or algorithm but uses a simulation of currents, and the learning rules are based on the integration of variables over time. To move towards such continuous version, an initial straightforward step would be to enable blended and smooth transitions between actions according to their activation. Or more sophisticated methods such as proximal policy optimisation (PPO) [72] for low-level motion control can be applied to the action outputs for continuous control tasks. A more significant alteration would be to move from discrete sensory states to a continuous (sparse) activation.

A potential application of this model is to provide milestones in a complex task for training a learning model. Directly training a model on a complex task is likely to be unstable in its early stages, because the reward is either not provided or has little correlation to actions. A

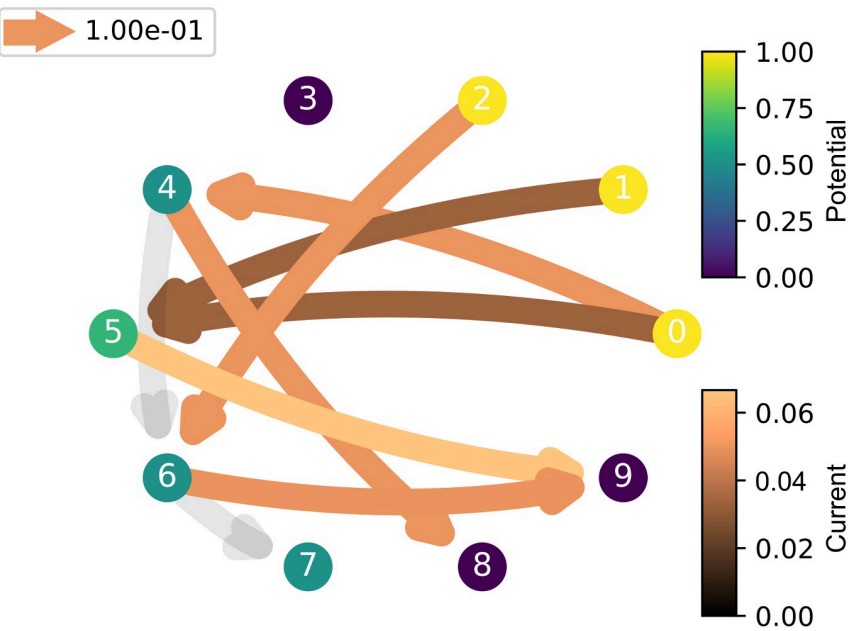

**Fig 13. The state network with a group of state nodes (0, 1, 2) activated and another group of state nodes (8, 9) set as a target.**

complex task can be decomposed into several easier tasks with their own goals, and rewards computed for these goals can have a better correlation to actions. These goals are subgoals of the original task, and they can be milestones guiding learning. Our model can be part of a framework in which the states are coded, causation is recorded, and subgoals are identified and ordered. Then the reward can be computed according to the currents and potentials. For example, a larger potential drop can indicate effective progress toward the final goal. In such a way, our model can be used as a critic in an actor-critic model.

## Methods

### State network and current flow

As shown in Fig 14A, a minimal state network consists of two state nodes $i$ and $j$. Each has two dimensions of activity: the injected current, which represents the state of the environment or agent by sparsely coding the sensory inputs; and the potential, which represents the target (the lower the potential, the stronger the target). There are directed connections between the state nodes which have associated weights $w_{i,j}$ acting as the conductance from state node $i$ to state node $j$. When there is a potential difference $V_{i,j}$, the current on the connection $i, j$ is:

$$I_{i,j} = \begin{cases} w_{i,j}V_{i,j} & \text{if } V_{i,j} > 0 \\ 0 & \text{if } V_{i,j} \leq 0 \end{cases} \qquad (4)$$

When there are multiple state nodes, every node is (potentially) connected to all other nodes forming a network of resistors. For convenience to calculate the potential and currents in the circuit at any time step, we can use circuit analysis approaches. Here, we use Nodal

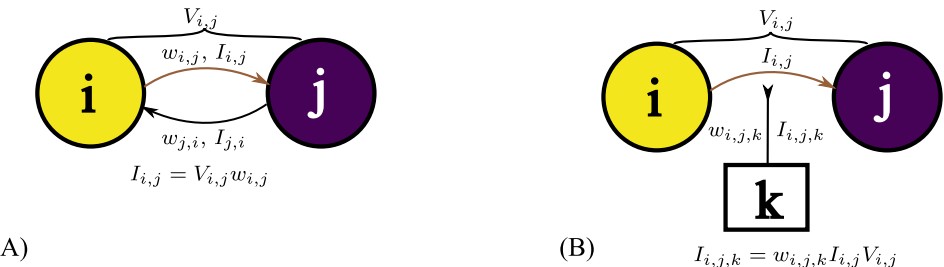

**Fig 14.** (A) A minimal state network: state node $i$ and state node $j$ have different potentials (note 'potential' here refers to potential in a resistive network, not the membrane potential of a neuron). The potential difference $V_{i,j}$ can cause current $I_{i,j}$ from $i$ to $j$ if there is a connection from $i$ to $j$ and $i$ has a higher potential than $j$. The weight of the connection $w_{i,j}$ is the conductance and the current $I_{i,j}$ follows Ohm's law. (B) A minimal circuit including an action node: an action node $k$ can be influenced by the state node connection it is attached to. The influence can be described as a function of a potential difference $V_{i,j}$, current $I_{i,j}$, and weight from the connection to the action node $w_{i,j,k}$. See text for details.

analysis, adapted to account for the use of unidirectional rather than bidirectional connections between nodes.

In this model, target states are set by pulling down the potential of the corresponding state node, by connecting the state node to the ground with some resistance. Lower resistance corresponds to a stronger goal; the exact value varies for different tasks or during learning, as specified in Table 6. The present state of the environment and agent activates a corresponding state node, whose potential is pulled up. Through nodal analysis we can then establish the resulting flow of current (see S1 Text). Note that, although we use the concepts of potential and current, the system could be interpreted in different terms, such as hard-to-compress fluid flows through the network to a sink, to serve a similar function. We note here that although this superficially resembles spreading activation (or breadth-first search) through a graph, the current flow between the present state and goal is more directed in a resistive net, as addressed in the discussion.

## Latent learning in the state network

We assume an initial very weak connection between all nodes in the network, which is modified by the experience of state transitions for an agent exploring an environment. In this implementation, we use one-hot coding, so only one state node actives at a given time step. The aim is to build a graph to represent the possible routes between states, in which the activated state node represents the present state, edges connecting the state nodes represent possible

**Table 6. Parameters used for each task.**

| Symbol | Variable | Taxi | Voronoi | Simple associative learning |
|---|---|---|---|---|
|  | max episode steps | 100000 | 10000 | 40 |
|  | number of states | 500 | 150 | 6 |
| $N$ | number of action nodes | 6 | 5 | 5 |
| $dt$ | time-step | 0.02 | 0.02 | 0.02 |
| $\alpha$ | learning rate | 0.05 | 0.05 | 0.001 |
| $\beta$ | learning rate ratio for second-order connections | 2 | 2 | 2 |
| $c$ | learning rate factor for second order synapse weight decrease | 400 | 400 | 400 |
| $w_{ij0}$ | initial connection strength | 0 | 0 | 0 |
| $w_{1max}$ | max connection strength | 1 | 1 | 10 |
| $w_{ijk0}$ | initial second order synapse weight | 0.01 | 0.01 | $1^{-10}$ |
| $w_{2max}$ | max ground conductance | 1 | 1 | 1 |

transitions between states, and the weight of an edge represents how often the transition happened previously.

Assuming weights have a maximum limit, if the model experienced state $\hat{i}$ and $\hat{j}$, the edge weight from node $\hat{i}$ to $\hat{j}$ is be updated by the following equation:

$$\frac{\mathrm{d}w_{i,j}}{\mathrm{d}t} = \alpha\left(w_{1\max} - w_{i,j}\right) \tag{5}$$

where $\alpha$ is learning rate, and $w_{1\max}$ is the maximum weight.

The outcome of this learning rule is that sequences of states that occur in the world will be encoded in stronger connections between the relevant state nodes. Thus the network connectivity will come to implement a model of the environment's state transitions.

In this model, there is no mechanism for reducing connection strength, but a simple approach would be to introduce some slow constant decay of all weights. This would provide some ability to adapt to changing environments, as state transitions that are no longer experienced would be gradually forgotten.

Note Eq 10 could potentially be generalised if we allowed a sparse encoding (rather than a one-hot encoding) of the state. If there are multiple state nodes active at the same time, the equation can be extended to alter the connections between the two sequentially active groups of state nodes. And although the simulated worlds we explore here are discrete, in a continuous world in which the activity of state nodes continuously changes, the connections could be built by finding negative correlations between state node activity changes, that is, the connections are built from state nodes with decreasing activity to those with increasing activity, as this is indicative of the transition from one state to another.

## Action decision

The action decision of the model depends on the currents and local potential difference between nodes. There are currents flowing through the edges and nodes in a graph if a target is set by grounding a state node, and the present activating state is set to a high potential. A key function of the circuit is to map a large range of state transitions into a small selection of actions. As shown in Fig 14B, for each connection between state node $i$ and state node $j$, there can be a 'second-order' connection to an action node $k$.

Assuming only one state node is active at each step we define the input to action node $k$ to depend only on the edge from this node that has the strongest power consumption:

$$\begin{aligned} I_k &= I_{i,j',k} \\ &= w_{i,j',k} I_{i,j'} V_{\hat{i},j'} \end{aligned} \tag{6}$$

where $\hat{i}$ is the index of the present state node, and $j' = \arg\max_j I_{i,j} V_{\hat{i},j}$, thus the edge between state node $\hat{i}$ and $j'$ is the edge with the maximum power consumption among the edges from node $\hat{i}$. This effectively sets a subgoal of the agent, i.e. the next state, $j'$, that it wants to reach to get nearer to the target state. It is possible for more than one action to be connected to this edge, in which case the probability of choosing an action depends on the currents of the action nodes:

$$P(a) = \frac{I}{\sum_{k=1}^{N} I_k} \tag{7}$$

where $a$ represents actions, $I$ is a vector containing the postsynaptic currents of each action node, $N$ is the number of action nodes. This could be interpreted as a lateral inhibition

function between the action nodes. We also explored the action probability after applying the softmax function, which is an approximation of lateral inhibition. Because of the normalisation, the largest input to softmax is 1 and the smallest input can be 0, softmax enlarges the probability of small inputs, resulting in more exploration. However in the following experiments we did not use softmax.

Note that as described, with only one edge contributing input, and only one action chosen, we can achieve the same probability for choosing the action using a simplified input calculation (because the power consumption terms cancel out in Eq 7), changing Eq 6 to:

$$
\begin{aligned}
I_k \quad &= I_{i,j',k} \\
&= w_{i,j',k}
\end{aligned}
\tag{8}
$$

That is, the choice between actions depends only on their relative weights. However, the requirement to consider only the edge with the strongest power consumption could be relaxed, allowing multiple edges from an active node to act as subgoals that contribute input to action nodes, which would have the form:

$$
\begin{aligned}
I_k \quad &= \sum_{j=0}^{N} I_{i,j,k} \\
&= \sum_{j=0}^{N} w_{i,j,k} I_{i,j} V_{\hat{i},j}
\end{aligned}
\tag{9}
$$

where $N$ is the number of state nodes. Then the power consumption $I_{i,j} V_{\hat{i},j}$ would represent the importance of state node $j$ as a subgoal, while $w_{i,j,k}$ represents how likely action node $k$ is to cause the transition from the present state to state node $j$. This could be further extended to the case where we use a sparse rather than one-hot encoding of the state, which would then additionally require summing over $\hat{i}$ for all active nodes.

Note that an MBON in a typical MB model outputs actions or behaviours, and its postsynaptic current depends on the activity of the KCs and synaptic weights between KCs and the MBON, but not the connections between KCs. A possible biological interpretation of Eq 6 is that the efficacy of the chemical synapse transmission from a KC axon to an MBON is modulated by the gap junction connections, and consequent current flow, from that KC axon to other KC axons. For example, an action potential in one KC could result in the creation of an action potential in the axon of the KC to which it is most strongly connected by a gap junction [3] resulting in neurotransmitter release from both synapses with the effects summed by the MBON.

## Setting a target in the state network

The state network builds a graph for the causality of actions and states by learning, while the motivation to execute the series of actions comes from reaching a target. In this model, the target state is set by pulling down the value of a state node in the state network, specifically, by increasing the conductance between the state node and the ground. In the sparse reinforcement tasks we explore, a reward indicates the accomplishment of the task. In the model, when reward is obtained, the conductance between the current state and ground is set to a fixed value (see Table 6 for parameters), marking a target state for future episodes.

$$
w_{i,\text{gnd}} = w_{2\text{max}}
\tag{10}
$$

With one target or reward state, the absolute values of current or potential are not important, given that the action-state route from the present state to the target state is found by the relative magnitude of the current and the local potential difference (Eq 6). However, it is also possible for the system to function when more than one target is set, as we explore in some examples, in which case the conductance value could influence the choice of target.

Note that in some of the tasks considered here, the target is explicitly provided by the environment, and changes from episode to episode. When a new target is provided to the model, the corresponding state is connected to the ground with a large conductance, and the previous target is released by setting the corresponding conductance to 0. In this type of task, the environment does not provide a reward value for training, and the model need not discover the target according to rewards.

## Exploration and exploitation

To learn the relationship between actions and state transitions requires the agent to explore the environment. Two factors influence exploration: action choice and subgoal selection.

Given a subgoal, which means an edge from the present state to the next state is selected, the choice of action is stochastic. The weights from the edge to action nodes are effectively parameters for the probability (Eqs 7 and 8). The chosen action might lead to the subgoal or a different state. No matter what happens, the learning rule will update the connections between the present state and the next actual state, as well as the weight between this edge and the actual action, as detailed below.

The subgoal selection is guided by currents and deterministic. In the edges from the present state, once there is an edge that has power consumption stronger than other edges, the edge is selected and the corresponding next state is the next subgoal. Hence, the currents make the agent concentrate exploration on the route most likely to be passable, effectively resulting in the exploitation of the information acquired so far by the network.

If all edges have equal power consumption (e.g., in the first episode) the subgoal selection is random. Stochastic selection (instead of deterministic) can optionally be introduced throughout the task by making the subgoal selection treat the power consumption on all edges from the present state as the likelihood to select the next state (instead of choosing the max edge). In this case, the agent will have a stronger tendency to explore, rather than exploit, the state space. Another (deterministic) way to introduce more exploration would be to connect unvisited states to ground, so that the current flow will tend to set up trajectories towards unvisited states. However, we found in practice this did not generalise well across different tasks, ending up with more limited exploration.

## Learning actions

With discrete actions, when a transition from state $\hat{i}$ to state $\hat{j}$ follows action $\hat{k}$, then second-order connection $\hat{i}, \hat{j}, \hat{k}$ is strengthened by:

$$\frac{\mathrm{d}w_{\hat{i},\hat{j},\hat{k}}}{\mathrm{d}t} = \alpha\beta \tag{11}$$

where $w_{\hat{i},\hat{j},\hat{k}}$ is the weight from the connection $\hat{i}, \hat{j}$ to action node $\hat{k}$, and $\beta$ is a factor to control the learning rate ratio.

If an action is performed following a current between states $\hat{i}$ and $j'$, but does not result in the state $j'$, the corresponding second-order connection weight decreases:

$$\frac{\mathrm{d}w_{i,j',\hat{k}}}{\mathrm{d}t} = -\alpha c w_{i,j',\hat{k}} \tag{12}$$

where $c$ is a factor with a typical value of 400, which is used in our experiments. This weakens the association between the action and the edge that was chosen when taking the action does not result in transition to the expected state.

Considering this rule in terms of MB function, the equivalent would be to assume that the relevant pair of KC inputs have their weights to the MBON strengthened, such that (as above) near simultaneous action potentials in their axons (caused by gap junction connections) are more likely to activate the MBON in future. The learning rule itself depends on activity in the MBON caused by one active state being followed by the second active state. An intriguing possibility is that recently observed back-propagating spikes in MBONs [73] could be involved in such a learning mechanism.

## Parameters

**Applying a continuous model to a discrete task.**    Our model is described mathematically as a continuous model, which provides generality when it is applied to or extended for continuous tasks. For initial evaluations, we adopted discrete tasks in our experiments. By assigning each discrete step a time step, we can use numeral ordinary differential equation solvers (ODEs) to match continuous variables to discrete steps. In our implementation, we simply adopted the Euler method. The typical time scale in our experiment is 0.02s for each step.

## Supporting information

**S1 Fig. The final synaptic strengths between state nodes and changes during learning of four maggots in four different training protocols.** (A) trained with 'AM+/OCT', tested with 'AON', (B) trained with 'AM/OCT+', tested with 'AON', (C) trained with 'AM+/OCT', tested with 'AOF', (D) trained with 'AM/OCT+', tested with 'AOF'. Left column: the final synaptic strengths. Right column: the change of synaptic strengths. Each line is one connection between two states, colour coded as in the legend. The red vertical lines marks the change of Petri dish. (PDF)

**S2 Fig. The change of synapse from edges to action nodes during learning of four maggots in four different training protocols.** (A) trained with 'AM+/OCT', tested with 'AON', (B) Trained with 'AM/OCT+', tested with 'AON', (C) Trained with 'AM+/OCT', tested with 'AOF', (D) Trained with 'AM/OCT+', tested with 'AOF'. Each line is one connection from edge to action node, colour coded as in the legend. The red vertical lines marks the change of Petri dish. Weights that did not change are omitted. (PDF)

**S1 Text. Nodal analysis for a circuit with bidirectional resistors.** (PDF)

## Author Contributions

**Conceptualization:** Tianqi Wei, Barbara Webb.

**Formal analysis:** Tianqi Wei.

**Funding acquisition:** Tianqi Wei, Barbara Webb.

**Investigation:** Tianqi Wei.

**Methodology:** Tianqi Wei, Barbara Webb.

**Project administration:** Tianqi Wei, Qinghai Guo, Barbara Webb.

**Resources:** Qinghai Guo.

**Software:** Tianqi Wei.

**Supervision:** Barbara Webb.

**Validation:** Tianqi Wei.

**Visualization:** Tianqi Wei, Barbara Webb.

**Writing – original draft:** Tianqi Wei, Barbara Webb.

**Writing – review & editing:** Tianqi Wei, Qinghai Guo, Barbara Webb.

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
