## [Decision Letter · Decision Letter 0]

31 May 2023

Dear Dr. Webb,

Thank you very much for submitting your manuscript "Learning with sparse reward in a network inspired by the insect mushroom body" for consideration at PLOS Computational Biology.

As with all papers reviewed by the journal, your manuscript was reviewed by members of the editorial board and by several independent reviewers. In light of the reviews (below this email), we would like to invite the resubmission of a significantly-revised version that takes into account the reviewers' comments.

We cannot make any decision about publication until we have seen the revised manuscript and your response to the reviewers' comments. Your revised manuscript is also likely to be sent to reviewers for further evaluation.

Sincerely,

Joseph Ayers, PhD

Academic Editor

PLOS Computational Biology

Lyle Graham

Section Editor

PLOS Computational Biology

Reviewer's Responses to Questions

**Comments to the Authors:**

Reviewer #1: In their manuscript "Learning with sparse reward in a network inspired by the insect mushroom body", Wei and co-authors describe computational work in which they investigate a learning algorithm in the area of model-based reinforcement learning. The algorithm utilises a resistor network with asymmetric resistances that are updated according to a local plasticity rule to eventually be a model of the state-space of an RL problem. Activation of nodes of the network represents states of an RL task and transistions (connections between states) are associated with "actions" - presumably the activation of the equivalent of MBONs. The algorithm is tested on the taxiv3 OpenAI task and a more complex task of similar nature developed by the authors.

While the paper is easy to follow in broad strokes, the description of the algorithmic details is hard to understand or contains errors. This makes it very difficult to assess the merits of the work.

For instance, I am unable to parse equation (2). What is the covariance of \\Delta \\gamma_i, \\Delta \\gamma_j if \\Delta \\gamma_i and \\Delta \\gamma_j are just the changes of the currents injected into state nodes i and j from time step t to t+1 of a discrete-time simulation?

In equation (6), what is the role of the argmax functions for \\hat{i} and \\hat{j} and how does this relate to equation (7)? In this context, what does "for a larger circuit" mean?

It is never explained - as far as I can see - what the actions actually do in terms of the simulation of the agent/ learning algorithm. There are some confusing references to that the system may take a state that the previous action might not lead to etc which are not clear. It is also not explained what is meant by the system "taking a state" though, reading between the lines, it appears to have something to do with the maximum voltage taken in the network of state nodes.

Another one, equation 8 contains k_{w_{i,j}} but the text below talks about gamma_k. And what is the intuition between the powers 1/2 in the learning rules?

In order to be understood, these model description issues need to be resolved.

One good approach to take in this respect would be to explain one variant of the model fully and in detail; then describe the extensions or simplifications in other variants - not mixing descriptions of all variants together at each level.

Besides the lacking clarity in the algorithm, there are a couple of other concerns.

First, the learning results are compared to Q learning in which the penalty term (small negative reifnorcement) during the trial has been removed. This breaks the ability of Q-learning to accomplish the task. The authors argue their algorithm does not require the negative reinforcements or trial resets and is hence superior. This doesn't, however, take into account that the new algorithm builds a model during an exploration phase whereas Q learning does not do such a thing. Generally, it is not clear how costly the exploration phase is and how that compares to other approaches, especially other model-based RL algorithms.

Lastly, the big elephant in the room is whether this is an algorithm inspired by the insect mushroom body or not. The authors toe a fine line of giving a few similarities of the proposed algorithm with known or recently discovered properties of the mushroom body. But then they wisely use language for the model that sets it apart from the terms used for describing the MB. I suppose that nobody can put in doubt that if the authors say that they were inspired by the mushroom body that it was so. However, is it a helpful assignation to call the model MB-inspired? The authors already point out that assuming potentially all-to-all asymmetric KC gap junctions seems like a stretch. However, there is a much more unrealistic element to their algorithm: actions (presumably MBON activations) are associated with the magnitude of the current flowing through an asymmetric gap junction between two Kenyon cells. The authors need to clarify how this could happen in the nervous system or discuss clearly how the algorithm is clearly not nervous-system-like in this respect and hence should not be interpreted as a model of a brain circuit.

Finally, regarding interest for the general public, while the idea of bio-inspired (or not) machine learning algorithms is clearly interesting to the general public, I feel that this paper would be too technical to be particularly suitable for public outreach.

Reviewer #2: This is an interesting manuscript that proposes a unique neural circuit model for reinforcement learning. The model is inspired by the insect mushroom body (MB), incorporating some recently found ultrastructural features. While I believe this study to be significant and inspiring for people in the field of both computational and experimental neuroscientists, I have several minor concerns regarding how the model relates to actual biology.

Strengths:

The presence of extensive KC-KC connections was discovered only recently, and there have been only one experimental study (Manoim et al., 2022) that addressed one of their functions. To my knowledge, the present manuscript is the first theoretical study to propose another potential benefit of those connections.

The assumptions used in this study (e.g. plasticity of KC-KC or KC-APL connections) provides a few hypotheses potentially testable by experimental biologists.

The “dynamic routing model” proposed in the manuscript shows superior performance than existing reinforcement learning models at least under some sparse reward conditions. The model also significantly differs from typical MB-based circuit models.

The manuscript is well written and accessible for experimental neuroscientists as well.

Weaknesses:

Although the model is inspired by the recently discovered features of the insect MB circuits, there are several assumptions that are not biologically plausible or oversimplified. While I fully understand that the authors do not intend to claim their model to be an accurate model of the MB, it may diminish the enthusiasm by biologists. Considering the following points could improve the manuscript in this regard.

One of the whole marks of the MB coding is a sparse coding of a large number of stimuli by KCs. The proposed model assumes that a single KC, or state node, responds to a single “world state”. What would happen to the performance of the model if a given state is represented by a sparse (or dense) population of KCs?

It is not clear what kind of synaptic transmission and plasticity the authors are assuming to take place at KC-MBON synapses, or “action node”. It would help if they can propose it using biological terms.

In line 221, the authors claim that their tasks are more difficult than the ones previously used in the MB models. I am curious to know how the authors’ model with lateral connections between KCs performs in those simpler associative learning tasks. This may be out of the scope of the manuscript, but perhaps the authors could comment on it in Discussion.

KC-KC chemical connections are currently considered inhibitory (Manoim et al., 2022). The model is perhaps rather inspired by the presence of gap junctions, but it is difficult imagine that gap junctions are as plastic as the model assumes. Even if they are, it is unclear how the excitation in one KC propagates through multiple KCs via such gap junctions. Perhaps, the authors could test what happens if KCs are not connected in an all-to-all fashion but communicate more locally.

Reviewer #3: Review is uploaded as an attachment.

**Have the authors made all data and (if applicable) computational code underlying the findings in their manuscript fully available?**

Reviewer #1: Yes

Reviewer #2: None

Reviewer #3: Yes

PLOS authors have the option to publish the peer review history of their article (what does this mean?). If published, this will include your full peer review and any attached files.

Reviewer #1: No

Reviewer #2: No

Reviewer #3: No
---

## [Decision Letter · Decision Letter 1]

31 Dec 2023

Dear Dr. Webb,

Thank you very much for submitting your manuscript "Learning with sparse reward in a network inspired by the insect mushroom body" for consideration at PLOS Computational Biology. As with all papers reviewed by the journal, your manuscript was reviewed by members of the editorial board and by several independent reviewers. The reviewers appreciated the attention to an important topic. Based on the reviews, we are likely to accept this manuscript for publication, providing that you modify the manuscript according to the review recommendations.

Sincerely,

Joseph Ayers, PhD

Academic Editor

PLOS Computational Biology

Lyle Graham

Section Editor

PLOS Computational Biology

Reviewer's Responses to Questions

**Comments to the Authors:**

Reviewer #1: In the last decade, there have been a number of works that used the insect mushroom body as an inspiration for investigating certain neural networks and, increasingly, also for proposing more abstract models of learning. However, after studying the review reports of other reviewers, the response of the authors and the revised manuscript I feel this manuscript is going too far. While the methods are still not presented well (more about this below), it is now becoming clearer how the model works. This highlights that the model is an algorithmic model, inspired by resistor networks. The attempts of linking the main mechanisms of the model to insect mushroom bodies are contrived. For instance, Kenyon cells are spiking neurons which is hard to reconcile with being nodes in a resistor network. What would it mean for a KC to be “pulled down” or “pulled up”? Gap junctions between KCs wouldn’t just create a continuous field of current flows like the nodes in the resistor network. For the case of the connection from the KCs to the MBONs, the proposed formula for the current isn’t even realistic for a resistor network (a “current” that is a function of a voltage and a current between two other elements (two KCs) of the network). The added sentence “A possible biological interpretation of Equ 6 is that the efficacy of the chemical synapse transmission from a KC to an MBON is modulated by the gap junction connections, and consequent current flow, from that KC to other KCs.” is not helping the situation as it is a highly unrealistic proposition in many ways, including because of the spiking nature of KCs. I would judge the likelihood that anything resembling the mode of operation of this model is going on in insect mushroom bodies as practically zero.

Is it still useful to be inspired by the insect mushroom bodies? I believe the answer is no. As an additional guide, I have inspected the listed assumptions:

Assumption 1 is about sparse representations, which have been found in insect MBs. However, the model uses one-hot encoding which is a very common concept in other areas, including machine learning. I would argue, proper one-hot encoding is therefore more inspired from elsewhere.

Assumption 2 is highly unrealistic as discussed above and clearly inspired by resistor networks, not MBs.

Assumption 3 is unrealistic, given what is known about the APL/GGN in insects and inconsistent with the spiking nature of KCs.

Assumption 4 seems possible in terms of that gap junctions can be asymmetric and plastic but the information about this is not coming from MBs, so it’s bio-inspired not MB-inspired.

Assumption 5 is unrealistic in terms of the role of the KC to KC current flow and otherwise just describes the idea of a read-out layer found in any neural network and in many brain models, i.e. wouldn’t be a particular MB feature.

Assumption 6 refers to existing connectivity patterns in the MB but the point that synapses can be plastic and unsupervised learning can take place isn’t in any way something that is particularly related to insect MBs or that MBs would be particularly known for.

In summary, none of the assumptions appear particularly MB-inspired and some of them are in contradiction with current knowledge. I don’t think this makes it useful to try linking this model to insect MBs.

As it is written, the title, abstract, and vast majority of the introduction are about insect mushroom bodies which are largely irrelevant to the model. I think this is very unhelpful and the changes made to the discussion are not solving this problem.

I would suggest removing insect mushroom bodies from title, abstract and introduction completely and instead discuss resistor network as an inspiration and the reinforcement learning literature that the results relate to as the main background.

The Methods section is still not sufficient for a knowledgeable researcher to reproduce the model and remains confusing and imprecise. This includes the following problems (without claim of completeness of this list):

1. Appendix A is explaining how Ohm’s law leads to a set of linear equations but doesn’t state how these are solved (I suspect a standard algorithm from some linear algebra software package?) – however, what does “In practice, the solution converges in a few steps, which is less than the number of states.” refer to?

2. L590: “by connecting the state node to the ground with some resistance”: what resistance? Is it always the same?

3. L592: “Through nodal analysis we can then establish the resulting flow of current and associate it to actions, as described in Appendix A”: delete “and associate it to actions,” as this is not related to either nodal analysis or Appendix A.

4. L604: “An activated state node represents the present state, and edges connecting the state nodes represent possible transitions between states”: This is interpretation and doesn’t belong in the methods. Furthermore, more generally, it remains unclear how things are done. Presumably there is an update cycle with steps including activating one state, nodal analysis, calculating a “subgoal”, calculating action probabilities, choosing one of the actions, determining what the next state is? This needs to be described somewhere.

5. L606: “The weight of an edge represents how often the transition happened previously”: Similarly, not a description of the model.

6. L609: “can be updated by”: What does this mean: is it updated by this equation or not?

7. Equation 5: This is an ODE, how does it relate to the update sequence (point 4 above)? I am guessing that the update cycle is just discrete times steps, so how is the ODE for KC connection weights integrated? What are the timescales? What is the value of wmax?

8. Throughout the Methods the references to other sections are broken, e.g. L615: “section and” … “in section .”

9. L623: “And although the simulated worlds we explore here are discrete, in a continuous world in which the activity of state nodes continuously changes, the connections could be built by finding negative correlations between state node activity changes”: Unclear but trying to read between the lines, did they mean positive correlations between state node activations (maybe with a delay) in a Hebbian sense? (Note, also applies to L536)

10. L634: “the input to connection k” -> “input to action node k”?

11. L644: " This could be interpreted as a lateral inhibition function between the action nodes”: While winner-take-all dynamics can be implemented with inhibition, a softmax operation is a very specific function unlikely to be easily related to simple mutual inhibition.

12. L646: “Note that as described, the operation in Equ 7 cancels out the term of power

consumption in Equ 6. In this case, Equ 6 is equivalent to: …” -> while the cancellation occurs in equation (7) this does not make the stated formula valid, i.e. equation (6) is not equivalent to equation (8) only because equation (7) does not depend on “the term of power consumption”.

13. L651: the generalisation to multiple “subgoals” seems interesting but the formula (9) still assumes one single state i^hat contrary to the text talking about several active states.

14. L673: “In the model, when a strong reward is released, the conductance between the current state and ground is set to a value that is much larger than the conductance among KCs, marking a target state for future episodes.”: to what value is it set?

15. Equation 11 and 13: please don’t use “k” both as a free index and a summation index

16. L721: “If the causality of task is deterministic or with little randomness, the following learning rule can be applied”: what is “the causality of task”? And what does it mean that the learning rule “can” be applied? Is it applied? Always? In certain tasks? At certain times in certain tasks?

17. Throughout, e.g. L725: “with a typical value of …”: What does this mean? Is the value as stated? Or is it different in the different experiments? If the latter, please state the value for each of the experiments separately.

18. Please also state the values for all other parameters such as learning rates alpha etc

Reviewer #2: The authors have fully addressed my previous comments.

Reviewer #3: Uploaded as an attachment

**Have the authors made all data and (if applicable) computational code underlying the findings in their manuscript fully available?**

Reviewer #1: Yes

Reviewer #2: None

Reviewer #3: None

PLOS authors have the option to publish the peer review history of their article (what does this mean?). If published, this will include your full peer review and any attached files.

Reviewer #1: No

Reviewer #2: No

Reviewer #3: No

Figure Files:

Data Requirements:

Reproducibility:

References:

---

## [Editor Report · Decision Letter 2]

17 Apr 2024

Dear Dr. Webb,

We are pleased to inform you that your manuscript 'Learning with sparse reward in a gap junction network inspired by the insect mushroom body' has been provisionally accepted for publication in PLOS Computational Biology.

Best regards,

Joseph Ayers, PhD

Academic Editor

PLOS Computational Biology

Lyle Graham

Section Editor

PLOS Computational Biology

---

## [Editor Report · Acceptance letter]

3 May 2024

PCOMPBIOL-D-23-00288R2 

Learning with sparse reward in a gap junction network inspired by the insect mushroom body

Dear Dr Webb,

I am pleased to inform you that your manuscript has been formally accepted for publication in PLOS Computational Biology. Your manuscript is now with our production department and you will be notified of the publication date in due course.

With kind regards,

Anita Estes
